# Genetic Relatedness, Antibiotic Resistance, and Effect of Silver Nanoparticle on Biofilm Formation by *Clostridium perfringens* Isolated from Chickens, Pigeons, Camels, and Human Consumers

**DOI:** 10.3390/vetsci9030109

**Published:** 2022-03-02

**Authors:** Heba A. Ahmed, Rasha M. El Bayomi, Rehab I. Hamed, Rasha A. Mohsen, Fatma A. El-Gohary, Ahmed A. Hefny, Eman Elkhawaga, Hala M. N. Tolba

**Affiliations:** 1Department of Zoonoses, Faculty of Veterinary Medicine, Zagazig University, Zagazig 44511, Egypt; 2Department of Food Control, Faculty of Veterinary Medicine, Zagazig University, Zagazig 44511, Egypt; rmazab_2010@yahoo.com; 3Reference Laboratory for Quality Control on Poultry Production, Department of Poultry Diseases, Animal Health Research Institute, Zagazig Branch, Agriculture Research Center (ARC), Zagazig 44516, Egypt; rehab.44444@yahoo.com; 4Department of Bacteriology, Animal Health Research Institute, Mansoura Branch, Agriculture Research Center (ARC), Mansoura 12618, Egypt; rasha.mohsen@hotmail.com; 5Department of Hygiene and Zoonoses, Faculty of Veterinary Medicine, Mansoura University, Mansoura 35516, Egypt; dr.fatmagohary@gmail.com; 6Veterinary Hospital, Faculty of Veterinary Medicine, Zagazig University, Zagazig 44511, Egypt; ahmed_vet8_2007@yahoo.com; 7Department of Food Hygiene, Animal Health Research Institute, Mansoura Branch, Agriculture Research Center (ARC), Mansoura 12618, Egypt; emanelkhawaga@yahoo.com; 8Department of Avian and Rabbit Medicine, Faculty of Veterinary Medicine, Zagazig University, Zagazig 44511, Egypt; moonfacem2000@yahoo.com

**Keywords:** *Clostridium perfringens*, toxinotyping, biofilm inhibition, silver nanoparticles, RAPD-PCR genotyping

## Abstract

In this study, we determined the prevalence and toxin types of antibiotic-resistant *Clostridium perfringens* in chicken, pigeons, camels, and humans. We investigated the inhibitory effects of AgNPs on biofilm formation ability of the isolates and the genetic relatedness of the isolates from various sources determined using RAPD-PCR. Fifty isolates were identified using PCR, and all the isolates were of type A. The *cpe* and *cpb*2 genes were detected in 12% and 56% of the isolates, respectively. The effect of AgNPs on biofilm production of six representative isolates indicated that at the highest concentration of AgNPs (100 µg/mL), the inhibition percentages were 80.8–82.8%. The RAPD-PCR patterns of the 50 *C. perfringens* isolates from various sources revealed 33 profiles and four clusters, and the discriminatory power of RAPD-PCR was high. Multidrug-resistant *C. perfringens* isolates are predominant in the study area. The inhibition of biofilm formation by *C. perfringens* isolates was dose-dependent, and RAPD-PCR is a promising method for studying the genetic relatedness between the isolates from various sources. This is the first report of AgNPs’ anti-biofilm activity against *C. perfringens* from chickens, pigeons, camels, and humans, to the best of our knowledge.

## 1. Introduction

*Clostridium perfringens* is a Gram-positive anaerobic pathogen that inhabits the intestine of various animal species and humans. Necrotic enteritis (NE) is considered the main lesion caused by the organism in chickens and pigeons. The disease is fatal, and the main clinical manifestations include anorexia, depression, decreased growth performance, and reduced feed efficiency, thus resulting in severe economic loss [1]. In domestic animals, such as camels, the organism causes enterotoxemia, diarrhea, and sudden death [2].

In humans, *C. perfringens* causes gas gangrene, food poisoning, and gastrointestinal illnesses, including sporadic diarrhea, nosocomial diarrheal diseases, and antibiotic-associated diarrhea due to consumption of contaminated food with enterotoxin-producing strains [3,4]. Therefore, the presence of *C. perfringens* with a high count (>10^6^ CFU/g feces) is an indicator of *C. perfringens* food poisoning [5,6].

Five genotypes (A–E) have been recognized for *C. perfringens* based on the four main toxins (alpha, beta, epsilon, and iota toxins). Two other toxin types (F and G) have been recorded [7]. The most prevalent toxin is alpha-toxin, produced by all *C. perfringens* types and encoded by the *cpa* gene [8]. *C. perfringens* also produce other toxins that contribute to food poisoning and gastroenteritis in animals and humans. These toxins include enterotoxin, beta2 toxin, and perfringolysin O, which are encoded by the *cpe*, *cpb*2, and *pfo* genes, respectively [9]. The main virulence factor implicated in human food poisoning is enterotoxin [10], while the beta2 toxin has been associated with enteric diseases in humans and animals. However, the toxin-associated gene has also been found in isolates recovered from obviously healthy animals [11].

The economic losses caused by *C. perfringens* can be reduced using antibiotics, such as chloramphenicol, metronidazole, tetracycline, ampicillin, and imipenem to control bacterial infection. However, this has caused a significant increase in *C. perfringens* resistance to lincomycin, tetracycline, and erythromycin [12].

Several reports have described enhanced antimicrobial resistance among anaerobes and thus reduced susceptibility of clinical isolates to therapy [13]. The reason for the increased resistance is the uncontrolled use of antibiotics in animal production as growth promoters. This causes the emergence of multiple drug-resistant (MDR) isolates and the transfer of antimicrobial residues from livestock to humans, posing harm to consumers [12]. Therefore, the World Health Organization recommended the replacement of antibiotics with alternative strategies for growth promotion [14].

Biofilms are surface-related bacterial communities formed due to the adhesion of bacteria to surfaces and subsequent production of extracellular polymeric substances (EPs) [15]. The ability of bacteria to cluster and attach to themselves and be embedded in a self-produced matrix has been reported. The biofilm matrix is formed of polysaccharides, DNA, and proteins [16]. This matrix protects the bacteria against body defense mechanisms, and the effect of antibiotics and disinfectants. The biofilm layer enables the microbe to cause different diseases. Thus, 65–80% of infections are estimated to be caused by biofilm-forming bacteria [17].

Due to the smaller size and higher surface area to volume ratio, nanoparticles (NPs) have been investigated for their antibacterial and anti-biofilm effects [18]. Recently, the role of nanoparticles has been documented to overcome the limitations of antibiotics in controlling infections. Silver nanoparticles (AgNPs) have a biocidal effect on various foodborne pathogens [19]. AgNPs have been proven to have the potential to inhibit multidrug-resistant bacterial isolates, including Clostridial species. However, the role of AgNPs as biofilm inhibitors has not been investigated for *C. perfringens* isolates.

The relatedness of *C. perfringens* isolates from various sources has been previously reported using many typing methods, such as phage typing, serotyping, plasmid profile typing, ribotyping, multi-locus sequence typing (MLST), pulse-field gel electrophoresis, and repetitive element PCR (rep-PCR). Unfortunately, most of these methods are not always available in microbiology laboratories, and are technically fastidious and time-consuming. However, randomly amplified polymorphic DNA (RAPD) is a rapid typing method that is suitable and easily applicable in laboratories and is available for epidemiological tracing of the sources of infection [20].

Scarce information is available on the relative occurrence of α-toxigenic and enterotoxigenic strains of *C. perfringens* in general populations and camels, particularly in Egypt. Therefore, our study addresses the following: (a) the prevalence of *C. perfringens* in intestinal and fecal samples from chickens, pigeons, camels, and human consumers; (b) the prevalence of *C. perfringens* in retail chicken and camel meat; (c) toxin typing of the isolates; (d) antibiotic resistance profiles of the isolates; (e) inhibitory effect of AgNPs on biofilm formation ability of the isolates; and (f) genetic relatedness of the isolates from various sources determined by RAPD-PCR.

## 2. Materials and Methods

### 2.1. Sampling

Intestinal contents from freshly slaughtered chickens (*n* = 50) were obtained from flocks suspected to be affected with clostridial infection. The birds from farms were admitted to the Clinic of Avian and Rabbit Medicine Department, Faculty of Veterinary Medicine, Zagazig University, Egypt. Birds admitted to the clinic were subjected to clinical and postmortem examination. Intestinal samples (jejunum and ileum) were obtained from 30 pigeons (Balady) aged from 4 to 8 weeks with a history of depression, growth retardation, dropping of wings, and diarrhea. Additionally, diarrheic feces from camels (*n* = 50) were obtained from Sharkia Governorate, Egypt. Retail meat samples were also obtained from chickens (*n* = 50) and camels (*n* = 50) from retail shops in the study area. From human consumers, diarrheic stool swabs (*n* = 100) were obtained from patients at the outpatient clinic at Al-Ahrar Hospital, Sharkia Governorate, Egypt. Informed verbal/written consent for participation in the study was obtained from all participants, and it was approved by the Committee of Animal Welfare and Research Ethics, Faculty of Veterinary Medicine, Zagazig University, Egypt (protocol no. 118/2019).

### 2.2. Bacteriological Examination

The intestinal contents were directly obtained in sterile cooked meat broth (CMB; TM MEDIA, Titan Biotech Ltd., Delhi ISO 9,001, India). We aseptically homogenized 25 g chicken meat samples in 225 mL CMB. The human stool samples were directly enriched in CMB tubes. The tubes were anaerobically incubated at 37 °C for 24 h in an anaerobic jar containing gas generating kits (anaeroGen, OXOID Ltd., Hampshire, UK) for enrichment. For isolation of *C. perfringens*, a loopful from the enriched cultures were streaked onto the surface of reinforced clostridial agar (CM0151, OXOID Ltd., Hampshire, UK), and then anaerobically incubated at 37 °C for 24–48 h in an anaerobic jar containing gas generating kits. Presumptive shiny, pin-headed, and translucent *C. perfringens* colonies were biochemically identified using hemolysis tests, Gram stains, sugar fermentation, lecithinase tests, nitrate reduction, and motility tests.

### 2.3. Molecular Identification

We used PCR for confirmation of the biochemically suspected isolates. Following the manufacturer’s instructions, the extraction of DNA was conducted for 50 isolates using the QIAamp DNA Mini kit (Qiagen GmbH, Hilden, Germany, Catalog no. 51304).

Typing of *C. perfringens* was conducted by primers specific for alpha (*cpa*), beta (*cpb*), epsilon (*etx*), and iota (*iap*) toxin-associated genes [21]. Moreover, primers for the amplification of beta2 (*cpb*2) [22] and enterotoxin-associated genes (*cpe*) were also used [10].

### 2.4. Antimicrobial Susceptibility Testing

The antibiotic susceptibility of the isolates from chicken, pigeons, camels, and human sources was determined with the broth micro-dilution method. The 14 antibiotics used were penicillin (PEN), ampicillin (AMP), amoxicillin (AMX), ampicillin-sulbactam (SAM), clindamycin (CLI), metronidazole (MTZ), vancomycin (VAN), imipenem (IPM), meropenem (MEM), chloramphenicol (CHL), tetracycline (TET), cefotaxime (CTX), cefoxitin (FOX), and ceftriaxone (CRO). The interpretation criteria for the antibiotics were based on EUCAST (2019), except for TET, CTX, FOX, and CRO, which were according to the Clinical and Laboratory Standards Institute (CLSI) guidelines [23]. The minimum inhibitory concentration (MIC) was determined by double-fold dilution of the antimicrobials (0.125–256 µg/mL) in Brucella broth (OXOID Ltd., Hampshire, UK), as recommended by CLSI guidelines (CLSI 2011). The dilution was conducted in sterile 96-well flat microplates (0.05 mL 2× antimicrobial/well) using fresh culture from overnight growth on blood agar. The culture was suspended in 5 mL sterile deionized water to achieve 0.5 McFarland turbidity, then 0.1 mL suspension was added to 11 mL Brucella broth. Next, 50 mL of thoroughly mixed suspension was transferred to individual wells of microplates. The MIC was read as the lowest antimicrobial concentration that inhibits visible bacterial growth after anaerobic incubation at 37 °C for 24 h. Growth control, broth sterility, and *C. perfringens* ATCC 19574 strain were included in each run to evaluate the method’s reliability. The multiple antibiotic resistance (MAR) index was determined as the ratio of the number of antibiotics to which *C. perfringens* isolates exhibited resistance to the number of drugs for which the isolates were examined [24]. Multidrug resistance (MDR) is defined as the resistance of an isolate to at least one agent in three or more antibiotic classes, while extensively drug-resistant isolates (XDR) are defined as isolates resistant to at least one agent in all but two or fewer antimicrobial categories [25].

### 2.5. Biofilm Formation

The microtiter plate method was employed in evaluating the ability of *C. perfringens* isolates to form a biofilm [26,27]. Colonies under investigation were incubated overnight on blood agar under anaerobic conditions. Each isolate was adjusted to match McFarland obesity tube No. 0.5 (1.5 × 10^8^ CFU/mL) in Brucella broth. Then, 20 mL were distributed in wells of microtiter plates with a flat bottom. Each sample was distributed in three wells containing 180 µL tryptic soy broth supplemented with 1% sterile glucose. The plates were then incubated at 44 °C for six days under anaerobic conditions. The medium was then discarded, and the wells were washed using 200 µL sterile phosphate-buffered saline (PBS) trice to remove non-adherent cells. After drying the plates for 45 min, each well was stained with 110 µL crystal violet solution (0.4%) for 45 min, followed by washing twice using 350 µL distilled water. De-staining was conducted by pipetting 200 µL 95% ethanol in each well for 45 min. Finally, 100 µL of the de-stained solution was transferred to the wells in new sterile microtiter plates, and the amount of crystal violet was measured using an ELISA reader (model: sunrise R4, serial no. 610000079) at OD_620_ nm after adjustment to zero of the negative control. The experiment was conducted in triplicate, and the data were represented as mean ± standard deviation. The cut-off value (ODc) was calculated using the formula: ODc = average OD of negative control + (3 × standard deviation of negative control). The OD of each isolate was obtained by the formula: OD = average OD of the isolate—ODc. The data obtained were used to classify the strains as non, weak, moderate, and strong biofilm producers according to the following equations [28]: non-biofilm producer = OD ≤ ODc; weak biofilm producer = ODc < OD ≤ 2 × ODc; moderate biofilm producer = 2 × ODc < OD ≤ 4 × ODc; strong biofilm producer = 4 × ODc < OD [29].

### 2.6. Anti-Biofilm Activity of AgNPs-H_2_O_2_

AgNPs-H_2_O_2_ (Top Superpower-vision) was provided as a commercial product by El-Delta Center for Nanosilver Technology, Mansoura, Egypt. The stock solution of the product comprised 45 nm silver nanoparticles (0.00004467 mL/liter) with hydrogen peroxide (50% liter) and natural herbs, i.e., mint (1 mL/liter), at a concentration of 5 mL/liter of water. Then, the product was diluted using Mueller–Hinton broth.

Out of 14 strong biofilm-producing isolates, six XDR representative isolates from each source were chosen for the biofilm inhibition experiment. The anti-biofilm activity of the AgNPs was determined qualitatively using the tube method [30]. In brief, 50 μL overnight culture of the targeted bacteria in LB broth was further diluted to adjust its turbidity according to 0.5 McFarland Standards (5 × 10^5^ CFU/mL). The suspension was added to the tubes containing 2 mL sterilized Brucella broth, and these tubes were incubated at 37 °C for 24 h under anaerobic conditions after adding different concentrations of AgNPs (25, 50, 75, and 100 μg/mL) in separate tubes. Negative control without the bacterial suspension and the positive control left without the addition of AgNPs were also included in the experiment. After incubation, the broth culture was decanted and washed twice with PBS. The inside of the tubes was stained with crystal violet dye (0.1%) for 30 min; the excess dye was decanted and gently washed off using deionized water. The tubes were dried, and biofilm formation ability was determined by observing a thin layer of blue film on the walls of the tubes. For quantitative estimation of biofilm formation, six representative isolates were chosen from strong biofilm producers; the microtiter plate assay was used [26,27]. In this method, 96-well microtiter plates were used. The wells were inoculated with 180 μL Brucella broth and 10 μL culture grown overnight and further diluted to adjust its final concentration to 5 × 10^5^ CFU/mL and 10 μL AgNPs (concentrations used were 0–100 μg/mL) and incubated at 37 °C for 24 h under anaerobic conditions [30]. Negative and positive controls were also used in the assay using sterile growth medium only and working solution, respectively. The experiment was conducted in triplicate to evaluate its reproducibility, and the values were expressed as the average of the three independent experiments. The percentage of biofilm inhibition was calculated using the following formula according to Kalishwaralal, BarathManiKanth [30]:$$1-\frac{\mathrm{OD}620\;\mathrm{of}\;\mathrm{cells}\;\mathrm{treated}\;\mathrm{with}\;\mathrm{AgNPs}}{\mathrm{OD}620\;\mathrm{of}\;\mathrm{non}\text{-}\mathrm{treated}\;\mathrm{control}}\times 100$$

### 2.7. Genotyping

The genetic relatedness between *C. perfringens* isolates from chickens, pigeons, camels, and human patients was evaluated using RAPD-PCR [31] as previously described. The fingerprinting data were transformed into a binary code depending on the presence or absence of each band. A dendrogram was generated by the unweighted pair group method with an arithmetic average (UPGMA) and Ward’s hierarchical clustering routine. Cluster analysis and dendrogram construction were conducted using SPSS, version 22 (IBM, Armonk, NY, USA, 2013). The discriminatory powers of both methods were measured using Simpson’s index of diversity (*D*), indicating the average probability that a typing system will assign a different type to two unrelated strains randomly sampled from a population. A *D*-value of more than 0.9 indicates good differentiation [32] (Hunter, 1990).

### 2.8. Data Analysis

Data were introduced into R software (R Core Team, 2019; version 3.5.3) for data visualization. The R package “Complex-Heatmap” was used to build a heatmap based on virulence genes, biofilm category, and antimicrobials tested in isolates from chicken, pigeon, camel, and human consumers [33].

## 3. Results

### 3.1. Postmortem Examination of Intestinal Samples from Birds

Examination of the intestine from chickens showed subclinical NE (Figure 1A,B). Multifocal pale foci of mucosal necrosis were observed from the serosa. The lumen of the intestine was filled with gas bubbles in the broiler. The intestine was filled with thick, brownish watery exudate in the jejunum (Figure 1C,D). Examination of the pigeon’s intestine exhibited no evidence of enteritis.

### 3.2. Prevalence of C. perfringens in the Examined Samples

Analysis of the obtained samples revealed the isolation of 50 (15.2%) *C. perfringens* out of the 330 examined samples (Table 1). A higher isolation rate was obtained from pigeon intestinal contents (66.7%). There was no statistically significant difference between the isolation of *C. perfringens* from intestinal content (20%) and meat of chickens (10%), *p* = 0.26, and the prevalence of *C. perfringens* from camels was insignificantly higher in diarrheic feces (18%) compared to raw camel meat (4%) (*p* = 0.06). In human samples, 4% were positive for *C. perfringens*.

### 3.3. Toxinotyping of C. perfringens Isolates

All isolates were of type A (positive for the *cpa* gene), and only isolates from camel feces (3, 33.3%), camel meat (1, 50%), and human stools (2, 50%) were positive for the *cpe* gene (Table 1, Figure 2). The *cpb*2 gene was identified in 28 (56%) of the isolates from all the sources. None of the isolates were found to harbor beta-, iota-, or epsilon-associated genes.

### 3.4. Antimicrobial Susceptibility Testing

Antimicrobial susceptibility testing of 50 *C. perfringens* isolates from different sources revealed the resistance of 82–94% of the isolates to penicillin, cefotaxime, cefoxitin, ceftriaxone, clindamycin, and chloramphenicol (Table 2, Figure 2). However, all isolates were susceptible to vancomycin, and 86% exhibited susceptibility to metronidazole. Moreover, 74% of each of the examined isolates were sensitive to ampicillin, amoxicillin, and ampicillin-sulbactam. Multiple drug resistance was observed in 92% of the isolates (46/50), and the MAR index ranged from 0.28 to 0.9, with an average of 0.63 (Table 3).

### 3.5. Biofilm Formation

*C. perfringens* isolates under investigation were all biofilm producers, of which 14 (28%) were strong biofilm producers with an average OD620 of 0.5037 ± 0.03. All strong biofilm-producing isolates were resistant to at least ten antimicrobials with an MAR index of 0.7 or more, and they were classified as XDR isolates (Table 3). Moderate biofilm producers encountered 60% (30/50) of the isolates, and they were MDR with MAR indices of 0.2–0.6.

### 3.6. Anti-Biofilm Activity of AgNPs-H_2_O_2_

The effect of AgNPs on biofilm production of six representative *C. perfringens* isolates was estimated by the qualitative tube method using various concentrations of AgNPs. Positive results were indicated by the presence of a thin layer of biofilms after staining with the dye. The isolates from chicken, camels, and humans required a concentration of 100 µg/mL to inhibit biofilm formation. In the case of the isolate from pigeons, no biofilm production was observed at 75 µg/mL concentration (Table 4).

The quantitative biofilm estimation was conducted using the microtiter plate method, and the inhibition percentage of biofilm production with different AgNP concentrations was investigated. The inhibition of biofilm formation by *C. perfringens* isolates was dose-dependent. At the highest concentration of AgNPs (100 µg/mL), the six examined isolates showed inhibition percentages of 80.8–82.8% (Figure 3 and Appendix A).

### 3.7. Genotyping

The RAPD-PCR patterns of the 50 *C. perfringens* isolates from different sources were investigated by a single amplification profile. The multiple DNA fragments ranged in size from 300 to 2700 bp. The primer sets manufactured 33 profiles (referred to as R1–R33). The discriminatory power of the RAPD-PCR was calculated using Simpson’s index of diversity, and the *D*-value was 0.9763, which indicated a high discriminatory power. The dendrogram analysis of the examined isolates (*n* = 50) showed four clusters (Figure 4). The similarity between the human, pigeon, and camel isolates in the same cluster was 100%.

## 4. Discussion

*C. perfringens* is known as a major public health risk, causing diseases in animals and humans; it also causes economic losses, especially in poultry flocks [34]. This study investigated the prevalence of *C. perfringens* in various sources of animal and human origin. Toxin typing, antibiotic resistance profile, genotyping, biofilm formation ability, and the effect of silver nanoparticles on biofilm were also evaluated. The source of chicken meat contamination is mainly the intestinal contents during slaughtering and processing [4]. Fifty *C. perfringens* isolates were recovered from all the examined sources, and all the isolates were of type A. In our study, the isolation rate of *C. perfringens* was higher in pigeon intestines (66.7%), followed by chicken intestines (20%). The high prevalence of *C. perfringens* in chickens and pigeons could be attributed to the colonization of the organism in the intestinal tract early in poultry life, even from hatchery [35].

A study in Egypt reported that 41.7% of broiler chickens were infected with *C. perfringens*, of which 35.4% were asymptomatic [36]. In China, 23.1% of live poultry from farms and markets was positive, while 13.6% of fresh chicken meat was positive [4]. In Jordan, 43.2% of broiler chicken flocks were positive [37]. *C. perfringens* in poultry has acute clinical or sub-clinical forms. The acute disease leads to a high mortality rate and accounts for 1% of deaths per day [38]. The subclinical form is characterized by damage to the intestinal mucosa, which results in decreased digestion and absorption, reduced weight gain, and increased feed conversion ratio [39]. Although healthy birds usually contain less than 10^5^ colony-forming units (CFU) of *C. perfringens* per gram of digesta, NE is produced due to the proliferation of pathogenic *C. perfringens* strains to reach 10^7^ and 10^9^ CFU per gram of digesta [40].

Regarding pigeons, scarce studies reported the prevalence of *C. perfringens*; for instance, Rahman, Sharma [41] reported an isolation rate of 33.3%. Despite the high isolation rate in pigeon intestine, no NE lesions were observed. However, the birds under investigation had symptoms of diarrhea and growth retardation. This could be explained by the presence of another microbial pathogen, which in turn decreased the immunity of the pigeons, thus causing the high isolation rate of *C. perfringens* [42].

All *C. perfringens* toxin types produce alpha-toxin; however, higher amounts of the toxin are produced by type A than the other types [43]. Necrotic enteritis and the subclinical form of *C. perfringens* infection in poultry are caused by *C. perfringens* type A, and to a lesser extent type C [44]. The predominance of type A from NE cases has been reported in different studies [4,37,45]. The main source of infection for chickens in farms and outlets is cross-infection via feces; therefore, proper hygienic measures, including disposal of contaminated bedding, are crucial to control *C. perfringens* infections in farms and markets [4].

None of the chicken or pigeon isolates harbored the *cpe* gene, whereas the *cpb*2 gene was identified in 73.3% (11/15) of isolates of chicken origin and 20% (4/20) of isolates of pigeon origin. Accordingly, Gharaibeh, A1 Rifai [37] reported the isolation of *C. perfringens* type A isolates from chicken flocks suffering from enteritis, and the isolates were not positive for enterotoxin-associated genes. In Jordan, all isolates recovered from broiler chickens with enteritis were classified as type A and non-enterotoxin producers [37]. Similar results were also reported in Finland [46] and Sweden [44]. Beta2 toxins have been associated with enteric diseases in humans and other animals and NE in birds [47]. This explains our findings where *cpb*2 gene was recovered from all examined sources with variable percentages.

In our study, *C. perfringens* type A was recovered from 10% of retail chicken meat samples. In Belgium, 100% of *C. perfringens* isolates from chicken meat were of type A [48], which suggests that *cpa* might be a universal gene in *C. perfringens* isolated from meat samples [49]. In Korea, 11.1% [50] and 33% [12] of retail chicken meat samples were positive for *C. perfringens*, while in Jordan, 43% of chicken meat samples were positive [37]. Chicken and ground chicken meat examined in Japan for *C. perfringens* contamination showed that 97% and 100% of the samples were positive, respectively [51]. The *cpe* gene, which encodes the enterotoxin, is reported in less than 5% of *C. perfringens* isolates [52]. All our chicken isolates were *cpe*-negative. In accordance, none of the isolates from chicken and beef meat samples were positive for other toxin encoding genes other than the *cpa* gene [12]. However, in the USA and Japan, only 1–3% of *C. perfringens* isolates from chicken harbored the *cpe* gene [10,53]. Another study in Japan reported 0.5–0.7% *cpe*-positive isolates from beef samples [10].

The high detection rate of the *cpa* gene in *C. perfringens* isolates could be attributable to the presence of the gene on the chromosome. While, *cpb*, *etx*, and *iap* are located on plasmids, the *cpe* could be found either on chromosomes or plasmids [7]. Therefore, acquisition or loss of plasmids might have a role in changing toxin types [49]. In our study, the absence of genes except *cpa* could be explained by the loss of mobile genetic elements, and this is following a previous study [12]. Furthermore, the carriage of the *cpe* gene was reported to vary; in food poisoning isolates, it is on the chromosome, while it is on the transferable plasmids in isolates from other gastrointestinal (GI) diseases, such as antibiotic-associated diarrhea [51].

The first report on beta2 toxin and its encoding gene (*cpb*2) was in *C. perfringens* type C isolates from necrotizing enterocolitis in a piglet [54]. No significant homologies were observed between the amino acid sequence of *cpb*2 and *cpb* from the beta toxin [55]. The beta2 toxin was found to possess a weaker cytotoxic activity than the beta toxin, despite having similar biological activity [54]. However, a possible association was reported between enteric disease and the presence of *C. perfringens* isolates carrying the *cpb**2* gene [56,57]. In this study, the *cpb*2 gene was identified from *C. perfringens* strains isolated from chickens, pigeons, camels, and human sources. Our results conform with previously published studies that identified the *cpb*2 gene in isolates from chickens [56,58], humans [47], and camels [58]. A recent study in Egypt documented a high frequency of the *cpb*2 gene from *C. perfringens* isolates in various sources, including chickens and humans [59]. In the USA, all isolates from chicken and human patients were of type A, and the *cpb*2 gene was isolated from 93% and 65% of chicken and human isolates, respectively [60]. An investigation of food poisoning outbreaks showed that undercooked meat contaminated with *C. perfringens* resulted in the survival and growth of the organism, which spread to other servings, causing food poisoning in consumers [61]. Nearly 70% of *C. perfringens* food poisoning outbreaks and 20% of all non-foodborne gastrointestinal diseases are caused by enterotoxigenic *C. perfringens* [5]. In Korea, 33% of chicken meat samples were positive for *C. perfringens* and all isolates were of type A, with positive results for only the *cpa* gene and harmful for the *cpe*, *cpb, etx, iap*, and *net*B genes [12].

Fewer studies have documented the isolation of *C. perfringens* from camels; in our study, 18% of diarrheic feces from camels were identified, and all nine isolates were of type A, of which seven (77.8%) and three (33.3%) were positive for *cpb*2 and *cpe* genes, respectively. According to Fayez, Elsohaby [62], the isolation of *C. perfringens* from diarrheic calves and adults was 73.7%, and most of the isolates were of type A. Moreover, other studies have reported that *C. perfringens* type A is the predominant type from camels and other animals [63,64]. Scarce information has been reported regarding the occurrence of *C. perfringens* in camel meat. Our results showed that two (4%) retail camel meat samples were contaminated, and one isolate was positive for the *cpe* gene, while both isolates harbored the *cpb*2 gene. These findings show the potential of camel meat to serve as a source of *C. perfringens*. The presence of *cpb*2 gene is correlated with gastrointestinal diseases in humans and other animals. The two isolates from camel meat in our study were XDR and strong biofilm producers, thus showing the potential of these isolates to cause foodborne gastroenteritis in human consumers. Previous studies have reported the isolation of *C. perfringens* from retail meat from various animals [10,65]. These results exhibit the urgent need for high sanitary conditions during the slaughter of animals and during the retail of meat at markets to minimize the risk of foodborne intoxication caused by *C. perfringens* [65].

Healthy people carry less than 10^5^
*C. perfringens* CFU/g feces, while diarrheic patients may carry 10^6^ or more CFU/g [66]. The intestinal carriage of *C. perfringens* may be diverse, and some healthy subjects may also act as potential reservoirs of more than one *C. perfringens* strain [6]. In this study, *C. perfringens* isolates were recovered from 4% of human stool from diarrheic patients and were positive for *cpa* and *cpb*2, while two were positive for the enterotoxin-associated *cpe* gene. In Ireland, *C. perfringens* was isolated from 7.6% of fecal samples from elderly subjects [67].

Yadav, Das [68] identified 43.8% *C. perfringens* type A isolates from human diarrheal cases, of which 49.1% were found to have only the *cpa* gene, and 35.1% were positive for *cpa* and *cpb*2. The diversity of the study population and the methods of detection could be the reason for the higher carriage of *C. perfringens* compared to this study [6].

*C. perfringens* isolates in this study showed resistance to penicillin, tetracycline, cefotaxime, cefoxitin, ceftriaxone, clindamycin, and chloramphenicol. This occurred following a study on *C. perfringens* from camels and herders; the findings indicated less effective antimicrobial activity of penicillin and cephems [62]. The use of penicillin and tetracycline as growth promotors and for the control of Gram-positive bacterial infections may explain the resistance reported in our study to these drugs [69]. In contrast, penicillin and tetracyclines exhibited sufficient activity against *C. perfringens* isolates from chicken and human isolates [37,59]. Clindamycin was reported to be effective against *C. perfringens* isolates [70]. However, our results showed a high resistance rate of 94%, which is comparable with other studies [71,72]. The low resistance of the isolates to ampicillin, amoxicillin, and ampicillin-sulbactam coincides with another study in Egypt [59]. *C. perfringens* isolates were resistant to amoxicillin, ampicillin, cefotaxime, tetracycline, and clindamycin at rates of 16.1%, 29.8%, 11.3%, 63.7%, and 70.8%, respectively [4]. Another study reported that the most resistant antimicrobial agent for the *C. perfringens* isolates was tetracycline (33/33, 100%), followed by imipenem (24/33, 72.7%), and 27 of the 33 strains (81.8%) were multiple drug-resistant, exhibiting resistance to at least three classes of antimicrobials [12].

Multidrug resistance was observed in 92% of the isolates, of which 26% were XDR, and the MAR indices ranged from 0.28 to 0.9 with an average of 0.63. These findings conform with previous studies that showed the wide spread of MDR and XDR *C. perfringens* isolates due to the uncontrolled use of these drugs as growth promoters in animals and for therapeutic uses in humans and animals, which has caused increased resistance [59,62]. It has been shown that MAR higher than 0.2 could be due to contamination from high-risk sources, such as humans and farm animals frequently exposed to antibiotics, thus resulting in potential risk to consumers [73].

Biofilm-forming microbes are responsible for causing 65–80% of infections because they can resist environmental and physical conditions and antibiotic treatment [74]. Although most studies focus on biofilm production and methods of reduction in single species, biofilms in nature mostly comprise multiple species, where inter-species interactions can affect the development, structure, and function of these communities [75]. These interactions could be in the forms of genetic and metabolic exchange and signaling to occur between microorganisms in biofilm communities; however, high throughput and high-resolution methods are needed to reveal these interactions [76]. Such interactions could be either competitive or cooperative, resulting in impacts on maturation, physiology, antimicrobial resistance, and virulence of these communities [76].

All our isolates were biofilm producers, and 28% produced strong biofilms, conforming with another study in Egypt [59]. Biofilms produced by *C. perfringens* isolates contribute to developing gastrointestinal diarrheal diseases in humans and animals due to the protective effect of biofilms against antibiotic treatment [26]. Several studies have reported the ability of *C. perfringens* isolates from different sources to produce biofilms [26,27,50,59]. Therefore, to control infections caused by biofilm-producing microbes, biofilm inhibitors should be used.

The chemical, biological, and physical complexity and dynamics of EPs resulted in poor susceptibility to antimicrobial agents [15]. The extracellular polymeric substances either bind to antimicrobials, delaying their diffusion, or they chemically react with antimicrobials, causing inactivation [77]. Therefore, nanoparticles have been used as alternatives for biofilm eradication. Nanoparticles play an important role as carriers of EPS matrix disruptors due to their intrinsic high surface area to volume ratio, providing a platform for the development of materials with a wide spectrum of mechanical, chemical, electrical, and magnetic properties [78].

The use of NPs in inhibiting biofilms has been previously documented for various bacteria [79,80,81]. This study indicated the potential of AgNPs to inhibit biofilm production at concentrations of 75 and 100 µg/mL, and the percentage of inhibition was 80.8–82.8% at the highest AgNP concentrations. The inhibitory effect of AgNPs was previously reported by Siddique, Aslam [81] on biofilm production by *Klebsiella pneumoniae* isolates. The results revealed 23–86% inhibition in the presence of different concentrations of AgNPs due to the disruption of the EPS matrix. Moreover, other studies reported 89% and 75% biofilm inhibition in *Staphylococcus aureus* and *Escherichia coli,* respectively [82]. Studies also demonstrated that nanosized silver (at the size of 100 nM) resulted in a 95% reduction in biofilm production by *Pseudomonas aeruginosa* due to the disruption of the EPS matrix [83].

Tracing the source and route of transmission of *C. perfringens* type A strains causing food poisoning is significant in epidemiological investigations. RAPD-PCR is one of the effective methods used for studying the genetic relationship of *C. perfringens* isolates from various sources [20]. Our results indicated that 50 *C. perfringens* isolates recovered from various sources were grouped in four clusters. One of the clusters included isolates from humans, pigeons, and camels with 100% similarity, showing a common source of the isolates. Similar results were reported by Afshari, Jamshidi [20], who obtained *C. perfringens* isolates from human stool, poultry meat, and minced meat in the same cluster. The discriminatory power of RAPD-PCR in our study was 0.9763, which was a high discrimination of the reaction to the isolates from various sources. Our isolates of chicken and pigeon origin were genetically clustered more closely to each other, and this was consistent with a previous study [84].

The control of biofilm-producing bacteria especially in the food chain depends mainly on the identification of their levels and tracing their sources. Bacterial biofilms possess high resistance to commercial antimicrobial agents, which necessitate the need for finding control alternatives. The inhibitory action of AgNPs-H_2_O_2_ on bacterial biofilms make it a promising wide-spectrum antibacterial agent. Because of the ecofriendly nature of the product, it can be safely used in the food industry, protecting human health. Further studies are required to find more antimicrobial alternatives for food safety.

## 5. Conclusions

In conclusion, MDR and XDR *C. perfringens* type A are predominant in chickens, pigeons, camels, and humans in the study area. The potential of the isolates to manufacture illness is predicted because of the presence of *cpe* and *cpb*2 genes. The anti-biofilm activity of AgNPs is described in the study against MDR and XDR *C. perfringens* isolates. This is the first report of AgNPs anti-biofilm activity against *C. perfringens* in chickens, pigeons, camels, and humans to the best of our knowledge. The RAPD-PCR method seems promising for the epidemiological investigation of foodborne diseases caused by *C. perfringens* and for contamination source tracking in the field of food hygiene and industry.

## Figures and Tables

**Figure 1 vetsci-09-00109-f001:**
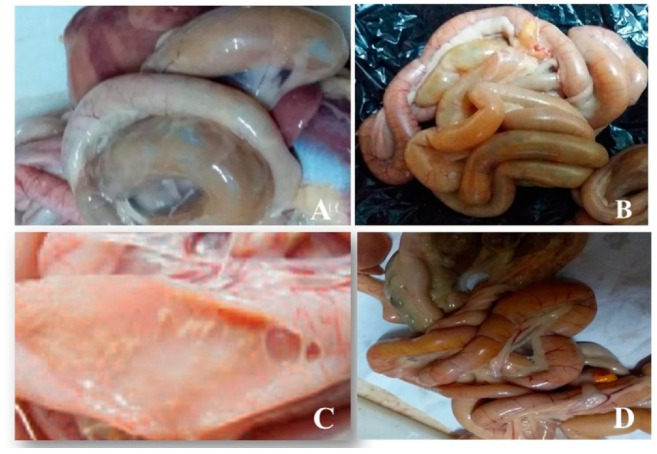
Multifocal pale foci of mucosal necrosis and the lumen of intestine filled with gas bubbles in broiler Hubbard chicken (**A**,**B**). Jejunum filled with thick, brownish watery exudate in cobb broiler chicken (**C**,**D**).

**Figure 2 vetsci-09-00109-f002:**
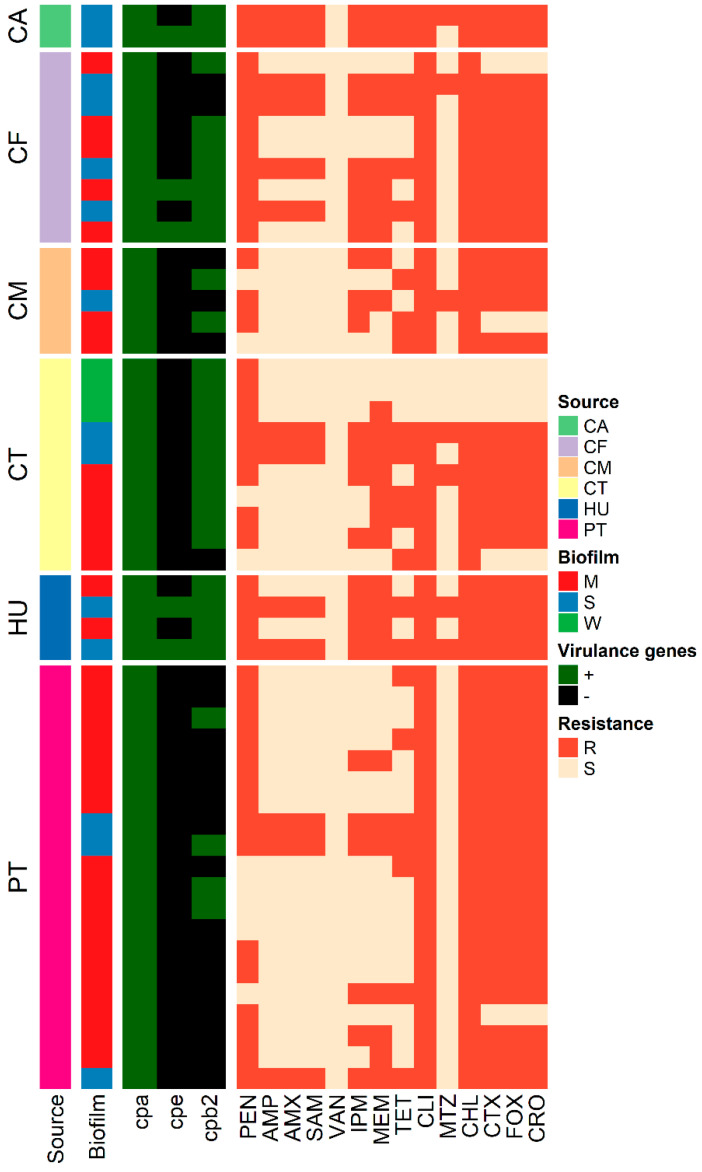
Heat map representation of virulence, biofilm, and antimicrobial resistance profiles of *C. perfringens* isolates recovered from chicken, pigeon, camels, and human consumers. CT: chicken intestine, CM: chicken meat, PT: pigeon intestine, CF: camel feces, CA: camel meat, HU: human stool, M: moderate, S: strong, W: weak, MAR: multiple antibiotic resistance. Penicillin (PEN), ampicillin (AMP), amoxicillin (AMX), ampicillin-sulbactam (SAM), clindamycin (CLI), metronidazole (MTZ), imipenem (IPM), meropenem (MEM), chloramphenicol (CHL), tetracycline (TET), cefotaxime (CTX), cefoxitin (FOX), and ceftriaxone (CRO). R: resistant, S: sensitive.

**Figure 3 vetsci-09-00109-f003:**
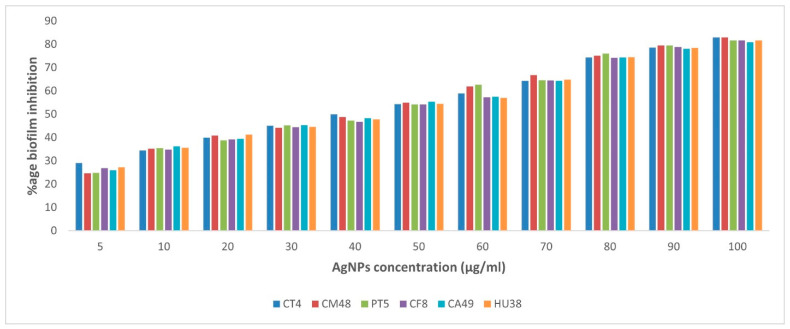
Percent inhibition of biofilm formation by various concentrations of AgNPs against *Clostridium perfringens*. The absorbance was measured at 620 nm for the quantification of biofilm formation. CT: chicken intestine, CM: chicken meat, PT: pigeon intestine, CF: camel feces, CA: camel meat, HU: human stool.

**Figure 4 vetsci-09-00109-f004:**
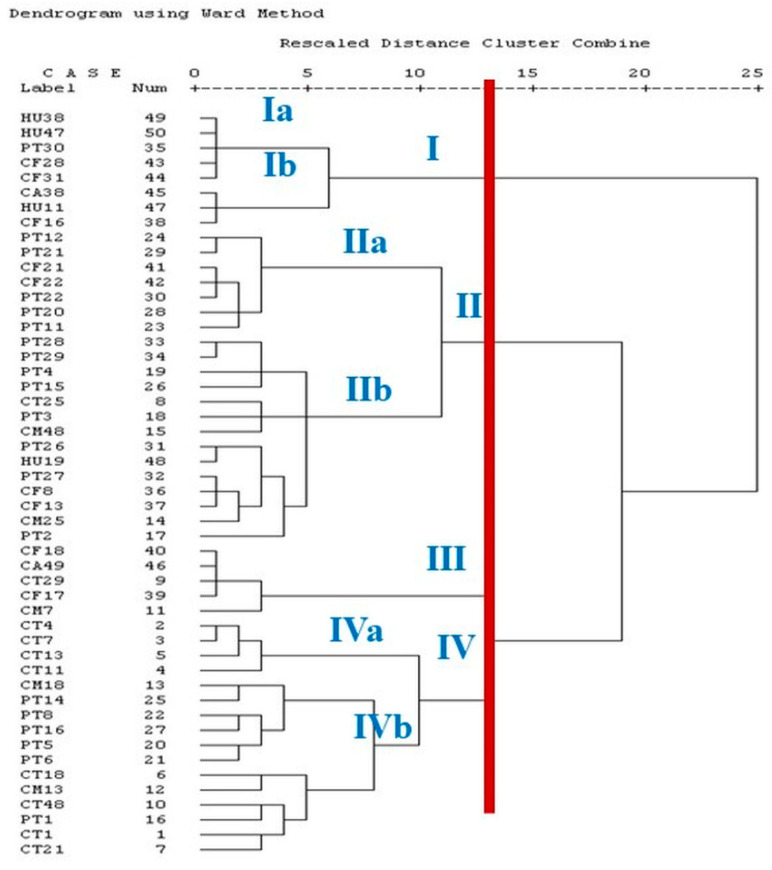
Dendrogram showing the relatedness of *C. perfringens* isolated from different sources as determined by RAPD-PCR fingerprinting using the SPSS computer software program. (CT: chicken intestine, CM: chicken meat, PT: pigeon intestine, CF: camel feces, CA: camel meat, HU: human stool).

**Table 1 vetsci-09-00109-t001:** Proportion and count of *Clostridium perfringens* isolates in chickens, pigeons, camels, and human samples.

Species	Type of Sample	Number Examined	Number Positive	Genotyping		
				*cpa* ^+^	*cpe* ^+^	*cpb*2^+^
**Chickens**	Intestinal content	50	10 (20%)	10	0	9 (90%)
	Meat	50	5 (10%)	5	0	2 (40%)
**Pigeons**	Intestinal content	30	20 (66.7%)	20	0	4 (20%)
**Camels**	Diarrheic feces	50	9 (18%)	9	3 (33.3%)	7 (77.8%)
	Meat	50	2 (4%)	2	1 (50%)	2 (100%)
**Humans**	Diarrheic stool	100	4 (4%)	4	2 (50%)	4 (100%)

*cpa*^+^, *cpe*^+^ and *cpb*2^+^ are calculated from the number positive.

**Table 2 vetsci-09-00109-t002:** Antibiotic susceptibility of 50 *Clostridium perfringens* isolates from chicken, pigeons, camels, and human sources.

Antibiotic Class	Antimicrobial Agent (Abbreviation)	S	R
**Penicillins**	Penicillin (PEN)	9 (18%)	41 (82%)
	Ampicillin (AMP)	37 (74%)	13 (26%)
**β-lactamas**	Amoxicillin (AMX)	37 (74%)	13 (26%)
	Ampicillin-sulbactam (SAM)	37 (74%)	13 (26%)
**Lincosamides**	Clindamycin (CLI)	3 (6%)	47 (94%)
**Nitroimidazole**	Metronidazole (MTZ)	43 (86%)	7 (14%)
**Glycopeptides**	Vancomycin (VAN)	50 (100%)	0
**Carbapenems**	Imipenem (IPM)	25 (50%)	25 (50%)
	Meropenem (MEM)	22 (44%)	28 (56%)
**Phenolics**	Chloramphenicole (CHL)	3 (6%)	47 (94%)
**Tetracyclines**	Tetracycline (TET)	27 (54%)	23 (46%)
**Cephems**	Cefotaxime (CTX)	7 (14%)	43 (86%)
	Cefoxitin (FOX)	7 (14%)	43 (86%)
	Ceftriaxone (CRO)	7 (14%)	43 (86%)

**Table 3 vetsci-09-00109-t003:** Toxin type, biofilm category, and antimicrobial resistance profiles of 50 *Clostridium perfringens* isolates from chicken, pigeons, camels, and human sources.

ID	Source	Virulence Profile			Biofilm Category	Resistance Pattern	MAR Index
		*cpa*	*cpe*	*cpb*2			
CT1	CT	+	-	+	W	PEN	-
CT4	CT	+	-	+	W	PEN	-
CT7	CT	+	-	+	W	PEN-MEM	-
CT11	CT	+	-	+	S	PEN-AMP-AMX-SAM-IPM-MEM-TET-CLI-MTZ-CHL-CTX-FOX-CRO *	0.9
CT13	CT	+	-	+	S	PEN-AMP-AMX-SAM-IPM-MEM-TET-CLI-CHL-CTX-FOX-CRO *	0.8
CT18	CT	+	-	+	M	PEN-IPM-MEM-CLI-MTZ-CHL-CTX-FOX-CRO	0.6
CT21	CT	+	-	+	M	MEM-CLI-CHL-CTX-FOX-CRO	0.4
CT25	CT	+	-	+	M	PEN-MEM-TET-CLI-CHL-CTX-FOX-CRO	0.5
CT29	CT	+	-	+	M	PEN-IPM-MEM-TET-CLI-CHL-CTX-FOX-CRO	0.6
CT48	CT	+	-	-	M	CLI-CHL	-
CM7	CM	+	-	-	M	PEN-IPM-MEM-TET-CLI-CTX-FOX-CRO	0.5
CM13	CM	+	-	+	M	CLI-CLI-CTX-FOX-CRO	0.3
CM18	CM	+	-	-	S	PEN-IPM-MEM-TET-CLI-MTZ-CHL-CTX-FOX-CRO *	0.7
CM25	CM	+	-	+	M	PEN-IPM-CLI-CHL	0.28
CM48	CM	+	-	-	M	TET-CLI-CHL-CTX-FOX-CRO	04
PT1	PT	+	-	-	M	PEN-TET-CLI-CHL-CTX-FOX-CRO	0.5
PT2	PT	+	-	-	M	PEN-TET-CLI-CHL-CTX-FOX-CRO	0.5
PT3	PT	+	-	+	M	PEN-CLI-CHL-CTX-FOX-CRO	0.4
PT4	PT	+	-	-	M	PEN-CLI-CHL-CTX-FOX-CRO	0.4
PT5	PT	+	-	-	M	PEN-IPM-MEM-TET-CLI-CHL-CTX-FOX-CRO	0.6
PT6	PT	+	-	-	M	PEN-CLI-CHL-CTX-FOX-CRO	0.4
PT8	PT	+	-	-	M	PEN-CLI-CHL-CTX-FOX-CRO	0.4
PT11	PT	+	-	-	S	(PEN-AMP-AMX-SAM-IPM-MEM-TET-CLI-CHL-CTX-FOX-CRO) *	0.8
PT12	PT	+	-	+	S	(PEN-AMP-AMX-SAM-IPM-MEM-TET-CLI-CHL-CTX-FOX-CRO) *	0.8
PT14	PT	+	-	-	M	TET-CLI-CHL-CTX-FOX-CRO	0.4
PT15	PT	+	-	+	M	CLI-CHL-CTX-FOX-CRO	0.3
PT16	PT	+	-	+	M	CLI-CHL-CTX-FOX-CRO	0.3
PT20	PT	+	-	-	M	CLI-CHL-CTX-FOX-CRO	0.3
PT21	PT	+	-	-	M	PEN-CLI-CHL-CTX-FOX-CRO	0.4
PT22	PT	+	-	-	M	PEN-CLI-CHL-CTX-FOX-CRO	0.4
PT26	PT	+	-	-	M	IPM-MEM-TET-CLI-CHL-CTX-FOX-CRO	0.57
PT27	PT	+	-	-	M	PEN-CLI-CHL	0.2
PT28	PT	+	-	-	M	PEN-IPM-MEM-CLI-CHL-CTX-FOX-CRO	0.57
PT29	PT	+	-	-	M	PEN-MEM-CLI-CHL-CTX-FOX-CRO	0.5
PT30	PT	+	-	-	S	(PEN-AMP-AMX-SAM-IPM-MEM-TET-CLI-CHL-CTX-FOX-CRO) *	0.8
CF8	CF	+	-	+	M	PEN-CLI-CHL	0.2
CF13	CF	+	-	-	S	(PEN-AMP-AMX-SAM-IPM-MEM-TET-CLI-MTZ-CHL-CTX-FOX-CRO) *	0.9
CF16	CF	+	-	-	S	(PEN-AMP-AMX-SAM-IPM-MEM-TET-CLI-CHL-CTX-FOX-CRO) *	0.8
CF17	CF	+	-	+	M	PEN-CLI-CHL-CTX-FOX-CRO	0.4
CF18	CF	+	-	+	M	PEN-CLI-CHL-CTX-FOX-CRO	0.4
CF21	CF	+	+	+	S	(PEN-AMP-AMX-SAM-IPM-MEM-TET-CLI-CHL-CTX-FOX-CRO) *	0.8
CF22	CF	+	-	+	M	PEN-IPM-MEM-CLI-CHL-CTX-FOX-CRO	0.57
CF28	CF	+	+	+	S	(PEN-AMP-AMX-SAM-IPM-MEM-TET-CLI-CHL-CTX-FOX-CRO) *	0.8
CF31	CF	+	+	+	M	PEN-IPM-MEM-CLI-CHL-CTX-FOX-CRO	0.57
CA38	CA	+	-	+	S	(PEN-AMP-AMX-SAM-IPM-MEM-TET-CLI-MTZ-CHL-CTX-FOX-CRO) *	0.9
CA49	CA	+	+	+	S	(PEN-AMP-AMX-SAM-IPM-MEM-TET-CLI-CHL-CTX-FOX-CRO) *	0.8
HU11	HU	+	-	+	M	PEN-IPM-MEM-CLI-CHL-CTX-FOX-CRO	0.57
HU19	HU	+	+	+	S	(PEN-AMP-AMX-SAM-IPM-MEM-TET-CLI-MTZ-CHL-CTX-FOX-CRO) *	0.9
HU38	HU	+	-	+	M	PEN-IPM-MEM-CLI-CHL-CTX-FOX-CRO	0.57
HU47	HU	+	+	+	S	(PEN-AMP-AMX-SAM-IPM-MEM-TET-CLI-MTZ-CHL-CTX-FOX-CRO) *	0.9

CT: chicken intestine, CM: chicken meat, PT: pigeon intestine, CF: camel feces, CA: camel meat, HU: human stool, M: moderate, S: strong, W: weak, MAR: multiple antibiotic resistance. Penicillin (PEN), ampicillin (AMP), amoxicillin (AMX), ampicillin-sulbactam (SAM), clindamycin (CLI), metronidazole (MTZ), imipenem (IPM), meropenem (MEM), chloramphenicol (CHL), tetracycline (TET), cefotaxime (CTX), cefoxitin (FOX), and ceftriaxone (CRO). * XDR isolates (extensively resistant isolates).

**Table 4 vetsci-09-00109-t004:** Qualitative estimation of color intensity of biofilm formation by tube method in absence and presence of silver nanoparticles (AgNPs) at different concentrations on *Clostridium perfringens* isolates.

Isolate	Positive Control	Negative Control	AgNP Concentrations (µg/mL)			
			25	50	75	100
**CT11**	++++	-	+++	++	+	-
**CM18**	++++	-	+++	++	+	-
**PT11**	++++	-	+++	+	-	-
**CF13**	++++	-	+++	++	+	-
**CA49**	++++	-	+++	++	+	-
**HU19**	++++	-	+++	++	+	-

CT: chicken intestine, CM: chicken meat, PT: pigeon intestine, CF: camel feces, CA: camel meat, HU: human stool.

## Data Availability

The data are included in the Appendix A.

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
