# Peer review of "Genetic Relatedness, Antibiotic Resistance, and Effect of Silver Nanoparticle on Biofilm Formation by Clostridium perfringens Isolated from Chickens, Pigeons, Camels, and Human Consumers"

_vetsci, 2022, doi:10.3390/vetsci9030109_

Round 1

Reviewer 1 Report

Improve the quality of the figures

Add at least 2 master tables in order to compare the results

Provide in-depth discussion

Add future perspective section

Author Response

We are pleased to resubmit a revised version of manuscript ID: vetsci-1597109 entitled " Genetic Relatedness, Antibiotic Resistance, and Effect of Silver Nanoparticle on Biofilm Formation by Clostridium perfringens isolated from Chickens, Pigeons, Camels, and Human Consumers" for Veterinary Sciences. Thank you for giving us the opportunity to revise and resubmit this manuscript. We appreciate the time each reviewer spent on this manuscript and the detailed comments that they have provided. The manuscript has been revised to reflect the reviewer's comments and suggestions. We look forward to working with you and the reviewers to move this manuscript closer to publication in Veterinary Sciences Journal.

We have responded specifically to each comment/suggestion below, the changes in the manuscript are indicated by Track Change option.  

Reviewer 1

-----------------

R1.1 Improve the quality of the figures

AU: We thank the reviewer for the comment. We checked the quality of the figures and we improved figures 1 and 4 as suggested.

R1.2 Add at least 2 master tables in order to compare the results

AU: We thank the reviewer for the comment. Two supplementary tables with the readings of the experiment are now included as suggested.

R1.3 Provide in-depth discussion

AU: We thank the reviewer for the comment. We revised the discussion as required.

R1.4 Add future perspective section

AU: We thank the reviewer for the comment. The following paragraph has been added at the end of the discussion section.

The control of biofilm producing bacteria especially in the food chain depends mainly on the identification of their levels and tracing their sources. Bacterial biofilms possess high resistance to commercial antimicrobial agents, which necessitate the need for finding control alternatives. The inhibitory action of AgNPs-H2O2 on bacterial biofilms made it a promising wide-spectrum antibacterial agent. Because of the ecofriendly nature of the product; it can be safely used in the food industry, protecting human health.  Further studies will be required to find more antimicrobial alternatives for food safety.

Reviewer 2 Report

The article presents the results of a comprehensive study of the epidemiological agent Clostridium perfringens isolated from various organisms and substances. The authors carried out a large amount of experimental research. Numerous isolates of the pathogen were used, which were investigated by various methods. Toxins production and the ability of biofilm formation by isolates of C. perfringens were studied. Also the resistance of C. perfringens to antibiotics was assessed. Special attention is paid to finding ways to suppress bacterial biofilms using a preparation based on silver nanoparticles. Wide range of methods including molecular genetic ones was used for solving the assigned tasks. The methods used are carefully described. The reliability of the obtained results is not in doubt. All parts of the work are important and interconnected. They make it possible to significantly expand the understanding of the circulation of C. perfringens and its potential danger to human and animals.

A few notes.

1) Keywords do not adequately reflect the content of the article.

2) Table 3 gives a lot of information about the results of testing isolates for resistance and sensitivity to antibiotics. However, the results of this table are poorly commented in the text, which makes it difficult to perception.

3) It should be clarified what underlies the choice of 6 C. perfringens isolates for experiments to test the effect of AgNP on biofilm formation.

4) In the discussion, the question of the possible interaction of C. perfringens with other microorganisms that may be part of biofilms was practically is not considered.

5) In the discussion more attention should be paid to the role of EPS in biofilm resistance to antibiotics. This is important, since the action of AgNP is aimed at disrupting the integration of the biofilm, which is associated precisely with the destruction of EPS.

6) It is desirable to emphasize the practical epidemiological significance of the work more clearly.

Author Response

We are pleased to resubmit a revised version of manuscript ID: vetsci-1597109 entitled " Genetic Relatedness, Antibiotic Resistance, and Effect of Silver Nanoparticle on Biofilm Formation by Clostridium perfringens isolated from Chickens, Pigeons, Camels, and Human Consumers" for Veterinary Sciences. Thank you for giving us the opportunity to revise and resubmit this manuscript. We appreciate the time each reviewer spent on this manuscript and the detailed comments that they have provided. The manuscript has been revised to reflect the reviewer's comments and suggestions. We look forward to working with you and the reviewers to move this manuscript closer to publication in Veterinary Sciences Journal.

We have responded specifically to each comment/suggestion below, the changes in the manuscript are indicated by Track Change option.  

R2.1 Keywords do not adequately reflect the content of the article.

AU: We thank the reviewer for the comment. We added keywords reflecting the content of the article.

Clostridium perfringens; Toxinotyping; Biofilm inhibition; Silver Nanoparticles; RAPD-PCR Genotyping.

R2.2 Table 3 gives a lot of information about the results of testing isolates for resistance and sensitivity to antibiotics. However, the results of this table are poorly commented in the text, which makes it difficult to perception.

AU: We thank the reviewer for the comment. We revised the discussion of Table 3 regarding the antibiotic resistance, the results are discussed in the text from lines 445-469.

R2.3 It should be clarified what underlies the choice of 6 C. perfringens isolates for experiments to test the effect of AgNP on biofilm formation.

AU: Thanks for the point. Out of 14 strong biofilm producing isolates, six XDR representative isolates from each source were chosen for the biofilm inhibition experiment. This is now included in the methodology and results sections.

R2.4 In the discussion, the question of the possible interaction of C. perfringens with other microorganisms that may be part of biofilms was practically is not considered.

AU: We thank the reviewer for the comment. The following paragraph has been added in the discussion section to cover this point.

Although most studies focus on biofilm production and methods of reduction in single species, biofilms in nature mostly comprise multiple species, where inter-species interactions can affect the development, structure and function of these communities (Burmølle et al., 2014). These interactions could be in the forms of genetic, metabolite exchange and signaling to occur between microorganisms in biofilm communities, however, this needs high throughput and high-resolution methods to reveal these interactions (Giaouris et al. 2015). This interaction could be either competitive or cooperative, resulting in an impact on maturation, physiology, antimicrobial resistance and virulence of these communities (Giaouris et al. 2015).

R2.5 In the discussion more attention should be paid to the role of EPS in biofilm resistance to antibiotics. This is important, since the action of AgNP is aimed at disrupting the integration of the biofilm, which is associated precisely with the destruction of EPS.

AU: Thanks for the point. The following paragraph has been added in the discussion section.

The chemical, biological, and physical complexity and dynamics of EPs resulted in poor susceptibility to antimicrobial agents. The extracellular polymeric substances either bind to antimicrobials, delaying their diffusion, or chemically react with them, causing inactivation (Anderl et al. 2000). Therefore, nanoparticles have been used as an alternative for biofilm eradication. Nanoparticles play an important role as carriers of EPs matrix disruptors due to  their intrinsic high surface-area-to-volume ratio, providing a platform for the development of materials with a wide spectrum of mechanical, chemical, electrical, and magnetic properties (Whitesides 2005).

R2.6 It is desirable to emphasize the practical epidemiological significance of the work more clearly.

AU: We thank the reviewer for the comment. The following paragraph has been added at the end of the discussion section.

The control of biofilm producing bacteria especially in the food chain depends mainly on the identification of their levels and tracing their sources. Bacterial biofilms possess high resistance to commercial antimicrobial agents, which necessitate the need for finding control alternatives. The inhibitory action of AgNPs-H2O2 on bacterial biofilms made it a promising wide-spectrum antibacterial agent. Because of the ecofriendly nature of the product; it can be safely used in the food industry, protecting human health.  Further studies will be required to find more antimicrobial alternatives for food safety.

Reviewer 3 Report

The paper by Ahmed et al is an excellent study on drug resistance in food animal guts and is highly timely. In an extension the authors have carried out a good study on the effects of biofilms formed by the bacteria involved.

The scientific work is broad and very well presented. I must commend the authors on the visual aspects of the work. The presentations are extremely clear.

A very nice piece of work that can be accepted as is.

Author Response

R3. A very nice piece of work that can be accepted as is.

AU: Authors would like to thank the reviewer for considering our manuscript a nice piece of work.

Round 2

Reviewer 1 Report

Accept